# Exhaled Breath Analysis for Investigating the Use of Inhaled Corticosteroids and Corticosteroid Responsiveness in Wheezing Preschool Children

**DOI:** 10.3390/jcm11175160

**Published:** 2022-08-31

**Authors:** Michiel A. G. E. Bannier, Sophie Kienhorst, Quirijn Jöbsis, Kim D. G. van de Kant, Frederik-Jan van Schooten, Agnieszka Smolinska, Edward Dompeling

**Affiliations:** 1Department of Paediatrics, Division of Paediatric Respiratory Medicine, School for Public Health and Primary Care (CAPHRI), Maastricht University Medical Centre+, 6229 HX Maastricht, The Netherlands; 2Department of Pharmacology and Toxicology, School of Nutrition and Translational Research in Metabolism (NUTRIM), Maastricht University, 6229 ER Maastricht, The Netherlands

**Keywords:** preschool wheezing, asthma, exhaled breath, volatile organic compounds, inhaled corticosteroids, treatment response, children

## Abstract

Exhaled breath analysis has great potential in diagnosing various respiratory and non-respiratory diseases. In this study, we investigated the influence of inhaled corticosteroids (ICS) on exhaled volatile organic compounds (VOCs) of wheezing preschool children. Furthermore, we assessed whether exhaled VOCs could predict a clinical steroid response in wheezing preschool children. We performed a crossover 8-week ICS trial, in which 147 children were included. Complete data were available for 89 children, of which 46 children were defined as steroid-responsive. Exhaled VOCs were measured by GC-*tof*-MS. Statistical analysis by means of Random Forest was used to investigate the effect of ICS on exhaled VOCs. A set of 20 VOCs could best discriminate between measurements before and after ICS treatment, with a sensitivity of 73% and specificity of 67% (area under ROC curve = 0.72). Most discriminative VOCs were branched C_11_H_24_, butanal, octanal, acetic acid and methylated pentane. Other VOCs predominantly included alkanes. Regularised multivariate analysis of variance (rMANOVA) was used to determine treatment response, which showed a significant effect between responders and non-responders (*p* < 0.01). These results show that ICS significantly altered the exhaled breath profiles of wheezing preschool children, irrespective of clinical treatment response. Furthermore, exhaled VOCs were capable of determining corticosteroid responsiveness in wheezing preschool children.

## 1. Introduction

Exhaled breath analysis has shown great potential in diagnosing various respiratory and non-respiratory diseases [1]. In children, volatile organic compounds (VOCs) in exhaled breath were able to differentiate wheezing preschool children and children with asthma from healthy controls [2,3]. Exhaled VOCs could also indicate asthma exacerbations at an early stage [4,5] and predict future asthma in wheezing preschool children [6].

Currently, two methods are being used for exhaled breath analysis: an “offline” method using gas chromatography mass spectrometry (GC-MS) to identify VOCs and an “online” method, which mainly comprises electronic nose (eNose) technology. Although both techniques have their own methodological and analytical strengths and limitations [1], an exhaled breath analysis by means of GC-MS is considered the golden standard technique. In addition to the different methods, many potential confounding factors exist, including diet, exercise and pharmacological treatment [7]. To date, the latter has been insufficiently investigated, and it is yet unclear to what extent medication influences exhaled VOCs’ patterns.

Moreover, exhaled VOCs might have the potential to predict a clinical corticosteroid response in wheezing preschool children. In adults with asthma, previous studies have shown the ability to predict oral steroid responsiveness [8] and loss-of-control after inhaled corticosteroid (ICS) withdrawal [9]. However, these studies have been performed in adults with confirmed asthma. The treatment of preschool wheezing has been debated for years, and it is yet unclear which wheezing preschool children might benefit from inhaled corticosteroids. Aeroallergen sensitisation and blood eosinophilia have been suggested as potential biomarkers to justify ICS prescription in wheezing preschool children [10]. However, in a recent randomised trial, blood eosinophilia-guided ICS prescription did not improve preschool wheezing management compared to standard care [11]. Hence, there is an urgent need for reliable biomarkers to predict steroid responsiveness in wheezing preschool children.

In the present study, we investigated the influence of ICS on the exhaled breath profiles of wheezing preschool children. We hypothesised that ICS significantly alter exhaled VOCs. Furthermore, we investigated whether exhaled VOCs could predict a clinical steroid response in wheezing preschool children.

## 2. Materials and Methods

### 2.1. Study Design

This study was part of the Asthma Detection and Monitoring (ADEM) study (clinicaltrial.gov: NCT 00422747), in which 202 wheezing children aged 2 to 4 years were prospectively followed until 6 years of age [12]. The primary goal of the ADEM study was to develop a non-invasive test for an early asthma diagnosis, including exhaled breath analysis and early lung function measurements. A detailed study protocol with in- and exclusion criteria was previously published [12,13].

Figure 1 shows the study flowchart of the current study. Children who experienced respiratory symptoms in the preceding month were included for a crossover ICS trial. If applicable, earlier prescribed ICS were stopped 4 weeks before the first study visit. During the initial visit, exhaled breath was collected, symptoms were assessed and lung function was measured (see details below). Children were instructed to refrain from solid foods and physical exercise 1 h before the measurement. Study visits were postponed in case of an airway infection. Atopy was determined by using the Phadiatop Infant test^®^ (Phadia, Uppsala, Sweden). Subsequently, children were randomly allocated to 8 weeks of ICS (Beclomethasone extrafine 100 μg twice daily, via the Aerochamber^®^, Teva Pharma NL, Haarlem, The Netherlands) followed by 8 weeks without ICS, or vice versa. No asthma medication other than salbutamol (Airomir^®^, Teva Pharma, Haarlem, The Netherlands, for symptom relief) was allowed during the study period. Compliance with the study medication was measured by weighing the ICS canisters before and after therapy. Children were excluded if less than 80% of the prescribed study medication was used. After 8 and 16 weeks, all measurements were repeated. Children were followed until the age of 6 years, when a clinical diagnosis (healthy, transient wheeze or true asthma) was made by two experienced paediatric pulmonologists and a computer-based algorithm [6].

### 2.2. Response to ICS

Children were considered responsive to ICS when there was a decrease in respiratory symptoms of ≥30% and/or an improvement in airway resistance of ≥10% [13].

Airway resistance was determined by using the interrupter technique (MicroRint^®^, Micro Medical Ltd., Rochester, UK). Respiratory symptoms were assessed by a short questionnaire evaluating symptoms of cough, wheezing and dyspnoea. A total symptom score before and after treatment was calculated, ranging from six points (most severe symptoms) to thirty points (no symptoms) [13].

### 2.3. Exhaled Breath Sampling and Measurements

Subjects were asked to breathe through a face mask connected to a valve of a resistance-free 1 L plastic bag (Tedlar bag, SKC Ltd., Dorset, UK). Within an hour of collection, the content of the sampling bag was transferred onto carbon-filled, stainless-steel desorption tubes (carbograph 1TD/Carbopack X, Markes International, Llantrisant, UK).

The exhaled breath samples were measured by means of GC-*tof*-MS, as described before [14]. The first steps consisted of releasing volatile compounds from the sorption tubes by thermal desorption using the Markes International Ultra-Unity automated thermal desorption equipment (Markes International). In the next step, 10% of the mixture of vapour was loaded onto a cold (5 °C) sorption trap, while the remaining 90% of the mixture was recollected into an identical sample tube. The vapour mixture was then reloaded from trap into the GC-*tof*-MS analysis. The temperature of the GC was programmed as follows: first, 40 °C for 5 min; then, increased by 10 °C every minute until 270 °C was reached. This temperature was maintained for 5 min. Electron ionisation at 70 eV was used with a 5 Hz scanning rate over a range of *m*/*z* 35–350.

### 2.4. Data Pre-Processing

The raw, original GC-*tof*-MS spectra were pre-processed before an actual statistical analysis. This consisted of noise removal, baseline correction and alignment, as previously described [14]. Each step of data pre-processing was performed to increase the quality of the data and to allow multivariate statistical analysis to focus on information of interest. For each peak in the total ion current chromatogram, the peak picking was performed, which consisted of calculating the area under the peaks by taking into account the corresponding mass spectra at specific retention time. Note that these areas are proportional to relative amounts of measured compounds. The absolute concentrations of the compounds were not determined. Finally, the area under the peaks for each sample was merged by combining the corresponding compounds based on retention time and similarities in mass spectra. To make the spectra comparable, the final step of pre-processing involved normalisation by probabilistic quotient normalisation.

The data were analysed by means of Random Forest (RF) and regularised multivariate analysis of variance (rMANOVA) [15]. RF was used to determine the effect of ICS on volatile metabolites in exhaled breath.

RF is a machine learning algorithm that creates many uncorrelated decision trees that predict samples into the appropriate class (here, pre-ICS and post-ICS). RF combines these decision trees to produce a general classification model. The RF model was validated using an internal test set, which was selected using the Kennard and Stone algorithm (20% of samples per class) [16]. This algorithm is based on Euclidian distance and allows for selecting a representative training set. The training set was used for optimisation steps (i.e., variable selection and selecting model complexity) and for developing a classification RF model. The test set was subsequently used to validate the constructed model. In the case of the RF model, an extra validation was employed using the so-called out-of-bag (oob) error. For each RF tree, one-third of the training samples was left out and not used in the construction of the classification model. These left-out cases were next used to establish the prediction error. Compound selection was based on the variable importance as assessed by RF in combination with an internal validation procedure in which the training set was iteratively split up into a training and validation set. This process was run using 1000 trees and 1000 iterations. The RF model provides a measure of the importance of a compound that gives the most important compound the highest value. Based on this value, a set of compounds that can discriminate between pre-ICS and post-ICS was selected. It is important to state that the discriminatory RF model was first constructed and optimised on a training set, containing 80% of samples of each group, and the final, optimised model containing only discriminatory VOCs was validated using an internal independent test set. Note that RF looks for patterns, i.e., sets of compounds that allow for predicting the class of interest, i.e., pre-ICS or post-ICS.

To visualise the results, a Principal Coordinate Analysis (PCoA) was performed on the proximity matrix obtained from the RF model. The model performance was demonstrated using receiver operating characteristics (ROC) curve. The statistical significance of responsiveness to the therapy was investigated by means of rMANOVA.

Lastly, the set of discriminating VOCs was putatively identified using the National Institute of Standards and Technology (NIST) library in combination with expert interpretation.

## 3. Results

Of 150 eligible children, 147 (98%) were able to stop taking ICS and were included in the trial. Unfortunately, only 65% of children appeared to be adherent to ICS treatment. Complete data were available for 89 children (Figure 1), of which 46 children were defined as steroid-responsive (52%). Subject baseline characteristics are given in Table 1. Baseline characteristics showed a clinically not relevant but statistically significant difference in airway resistance between responders and non-responders. The other characteristics were not significantly different between groups.

### 3.1. Influence of ICS on Exhaled VOCs

To identify the VOCs’ pattern related to ICS, the RF model compared individuals before and after ICS treatment. The RF model was first optimised on the training set and consequently validated using the test set. A set of 20 VOCs was selected as being the most discriminative. The final RF model, built on the data containing individuals before and after ICS treatment, and the set of 20 discriminative VOCs yielded a sensitivity and specificity of 73% (95% confidence interval 64–82%) and 67% (95% confidence interval 57–77%) for internal test samples, respectively. The corresponding ROC curve is shown in Figure 2.

The groups’ visualisation is shown in Figure 3A,B, where PCoA score plots are shown for training and projected test samples. The difference between pre- and post-ICS samples is observed between the first three principal coordinates (PCos), which explains 43.4% of the variance. The test samples are projected into the space of the corresponding training samples, which confirms the sensitivity and specificity of the RF model.

In Appendix A
Figure A1, responsiveness to ICS treatment is visualised in the PCoA score plots. This figure demonstrates no natural clustering with respect to clinical treatment response.

The top five compounds contributing to the discrimination between the pre- and post-ICS treatments were putatively identified using the NIST library (Table 2). Interestingly, the levels of four compounds were reduced after ICS treatment and the level of only one compound was increased after ICS treatment. The remaining VOCs were identified only in a general way as the following family of compounds: two terpenes, seven alkanes and four branched alkanes. The remaining two compounds could not be identified due to insufficient mass spectrum, overlap in the retention time or absence of mass spectrum in the NIST library.

### 3.2. Exhaled VOCs and Clinical Response to ICS

Due to the limited number of subjects, it was not feasible to create a reliable prediction model to discriminate between ICS responders and non-responders. Moreover, the study design included both children later diagnosed as transient wheezers and children later diagnosed as true asthmatics, for which treatment response might be different. As a result, discriminatory models for each group need to be built separately. Therefore, rMANOVA was used to demonstrate the usefulness of exhaled VOCs in determining steroid responsiveness. In this multivariate model, definitive diagnosis at an age of 6 years, i.e., true asthma or transient wheeze, as well as responsiveness to ICS were used as the main factors. Since airway resistance at baseline was statistically significantly different between responders and non-responders, it was included in the rMANOVA model as a potential confounder. The rMANOVA analysis yielded significant differences in VOCs’ profiles for responders and non-responders and for later-diagnosed true asthma and transient wheeze as individual factors, as well as the interaction between them with a *p*-value < 0.01. This significant interaction effect indicates that the variation is dependent on both factors simultaneously, and thus, the commonly encountered One-Variable at a Time approach for investigating treatment responsiveness would not necessarily result in the best overall optimal outcomes.

## 4. Discussion

This study showed that exhaled VOCs in wheezing preschool children were significantly influenced by an 8-week trial of ICS, irrespective of a clinical treatment response. Our classification model showed that 20 VOCs were the most discriminative, with a sensitivity of 73% and specificity of 67%. The majority of these VOCs were alkanes, but the top five discriminative VOCs (Table 2) also included aldehydes. Furthermore, exhaled VOCs before the start of ICS treatment were significantly different between wheezing children who were later classified as ICS-responsive compared to those classified as non-responsive. To the best of our knowledge, this is the first study investigating ICS-induced changes in exhaled VOCs in children, as determined by GC-MS. Moreover, this is also the first study that used exhaled VOCs to assess a clinical ICS response in wheezing preschool children.

Our findings significantly contribute to the knowledge of clinically applying exhaled breath analysis. Despite the 2017 ERS Task Force statement on exhaled biomarkers in lung disease [7], exhaled breath sampling and analysis is still not standardised and large heterogeneity between studies exists. One important factor in standardising breath sampling is the potential confounding role of pharmacological treatment. Only a few studies have focused on this topic. The study by Gaugg et al. showed significantly altered breath prints in adults, as assessed by SESI-HRMS, 10 and 30 min after inhalation of salbutamol, which was not shown after inhalation of the placebo [17]. However, in most studies, subjects are instructed to not use their medication 2 to 4 h prior to breath sampling. Unfortunately, that time window was not used in the study by Gaugg et al.; however, Brinkman and colleagues investigated the association of exhaled VOCs, as measured by GC-*tof*-MS, with urinary levels of salbutamol and oral corticosteroids in adults with severe asthma [18]. Their study showed that exhaled breath profiles can be linked to recent medication use, with fairly good accuracies. However, an important limitation of this study was that the specific time and dose of medication intake were uncertain.

In children, no similar studies have been performed to date. In a previous study from our research group, performed in the same study population, markers in exhaled breath condensate (EBC) showed no differences before and after ICS treatment [13]. Two other studies from the same research group, performed a metabolomic analysis of EBC in children with asthma [19,20]. In non-severe asthma, this method was not able to detect differences between steroid-naïve children and children using regular ICS [19]. Furthermore, EBC profiles showed no differences after a 3-week course of ICS, despite significant clinical improvements [20]. Probably, as also suggested by the authors, their approach was not capable of detecting the subtle differences in exhaled breath profile potentially caused by ICS.

By using GC-*tof*-MS, we were able to detect 20 VOCs that were most discriminative for ICS use, of which the top 5 discriminative VOCs are shown in Table 2. The majority of these identified VOCs were alkanes. Exhaled alkanes and, to a lesser extent, exhaled aldehydes have been identified as significant markers for airway inflammation in various asthma studies [21]. Interestingly, four out of our top five most discriminative VOCs, of which two are alkanes and two are aldehydes, decreased after 8 weeks ICS. These findings might suggest that the changes found in breath profiles were caused by the down-regulation of airway inflammation. However, we cannot exclude that the differences in exhaled VOCs were solely and directly induced by ICS, as we did not investigate drug levels or metabolites in blood and urine to compare our findings. Moreover, the ICS-induced changes in breath profiles were irrespective of a clinical treatment response, as shown in Figure A1. Of interest, in the study by Brinkman et al., octanal in exhaled breath was associated with urinary levels of oral steroids [18]. In our study, octanal was identified at the baseline visit, when subjects were steroid-naïve for 4 weeks. Moreover, after the 8-week ICS trial, octanal levels reduced, which implies that octanal is not a biomarker for ICS use.

In summary, we cannot fully conclude whether the ICS-induced changes in exhaled VOCs were directly caused by the drugs or whether it truly reflected anti-inflammatory changes in wheezing children. Nevertheless, our findings indicate that pharmacological treatment should be taken into account when performing breath analysis. Most importantly, when breathomics is used for diagnostic purposes, timing and dose of medication could be important confounding factors as it is unclear how long the metabolic effects persist. The common prohibition to use drugs 2–4 h prior to exhaled breath sampling might not be sufficient to reduce the impact on exhaled VOCs. In our study, we were fairly sure about medication use as we only included children in the final analysis when they were both steroid-naïve at inclusion and considered treatment-adherent after the trial. As mentioned before, exact medication intake was uncertain in the study by Brinkman et al. [18], and the effects of salbutamol were only investigated until 30 min after inhalation in the study by Gaugg et al. [17]. An extension of the latter study, including a longer follow-up duration and the use of various drugs, is of pivotal importance to improving the standardisation and reliability of exhaled breath analysis.

Another research aim of this study was the prediction of a clinical response to ICS in wheezing preschool children. Unfortunately, due to the high number of insufficiently adherent children, the sample size was too small to create a prediction model that could be properly validated. Nonetheless, our results showed significant differences in exhaled VOC profiles between ICS responsive and non-responsive wheezing preschool children. Moreover, this effect was dependent on the future diagnosis of these children at an age of 6 years: transient wheezing or true asthma. These results suggest that wheezing preschool children who eventually developed asthma had a higher chance of being steroid-responsive. Naturally, these results need validation and replication. However, the prediction of ICS responsiveness could be a unique and innovative potential application of exhaled breath analysis, as the treatment of wheezing preschool children has been debated for years [22]. Our findings are in line with a previous study in adults with asthma, in which an eNose was able to predict a clinical corticosteroid response with a good accuracy [8]. In children, most studies have focused on the use of fractional exhaled nitric oxide (FeNO) to predict a treatment response, with mixed findings [23]. In an earlier part of the ADEM study, we showed that FeNO and inflammatory markers in EBC could not predict a steroid response in wheezing preschool children [13]. Finally, in the study by Cavaleiro Rufo et al., an eNose analysis of EBC in children was able to identify children with asthma in need of ICS therapy [24]. Response to treatment was not investigated in this study. Our findings could implicate that exhaled breath analysis is able to better guide clinicians in identifying which wheezing children might benefit from ICS, potentially preventing both overtreatment and undertreatment with ICS and thereby reducing side effects as well as healthcare costs. The potential of exhaled VOCs as a diagnostic tool for ‘personalised medicine’ in this age group is highly interesting.

This study has several strengths. To the best of our knowledge, this is the first study to investigate the effect of ICS maintenance therapy on exhaled VOCs’ profiles in children at preschool age. Furthermore, this is the first study using exhaled VOCs, as measured by GC-MS, in determining clinical steroid responsiveness in preschool children. Our study was performed in a real-life setting, which makes it highly suitable for clinical practice. Moreover, our subjects were clinically well characterised, including details on medication use and treatment response.

Our study also had some limitations. First, weighing inhaler canisters is considered a suboptimal strategy to identify treatment adherence [25]. However, in this real-life clinical setting, we considered it to be the most optimal strategy. Second, as we were very strict in identifying adherence, 35% of participating children were excluded from analysis. As a result, the sample size was too small to create a reliable prediction model of treatment response. Another limitation is that we did not assess blood eosinophils in this study. However, we did measure FeNO, which is related to airway eosinophilia. In a previous study, FeNO could not predict a steroid response in the same population [13]. Finally, we did not investigate drug levels or metabolites in blood or urine to compare our findings.

## 5. Conclusions

An 8-week trial of ICS significantly influenced the exhaled breath profiles of wheezing preschool children, irrespective of a clinical treatment response. Moreover, exhaled VOCs were capable of determining corticosteroid responsiveness in wheezing preschool children. These results highlight the urgent need for further standardisation of exhaled breath analysis and its potential to guide therapy in a ‘personalised medicine’ approach.

## Figures and Tables

**Figure 1 jcm-11-05160-f001:**
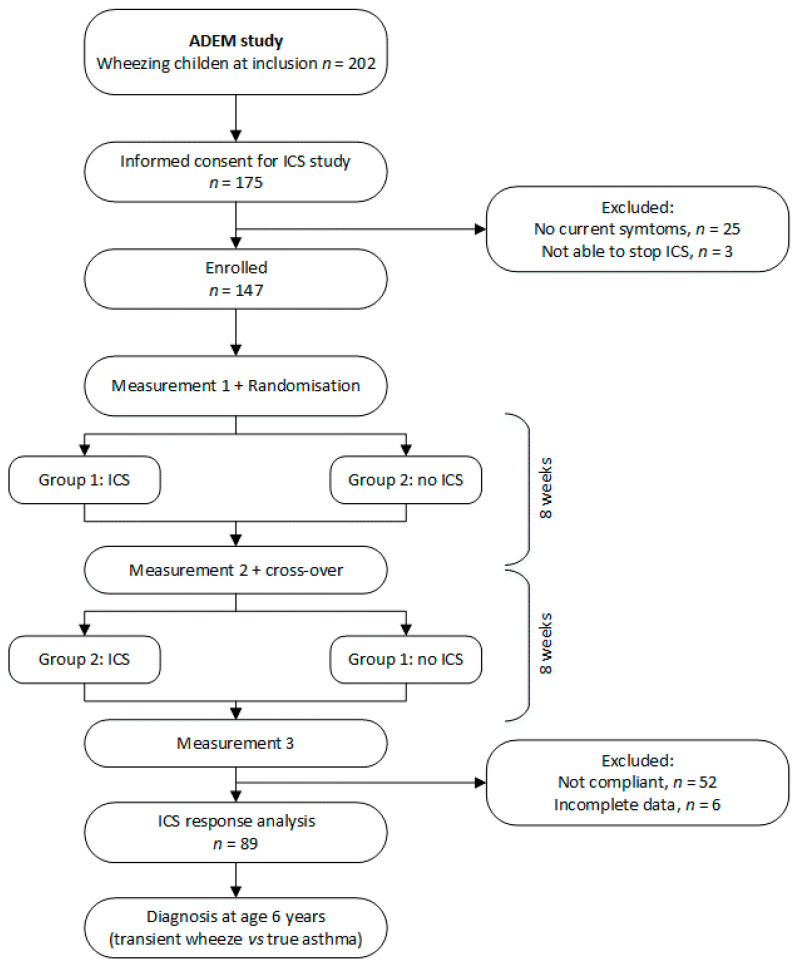
Study flowchart.

**Figure 2 jcm-11-05160-f002:**
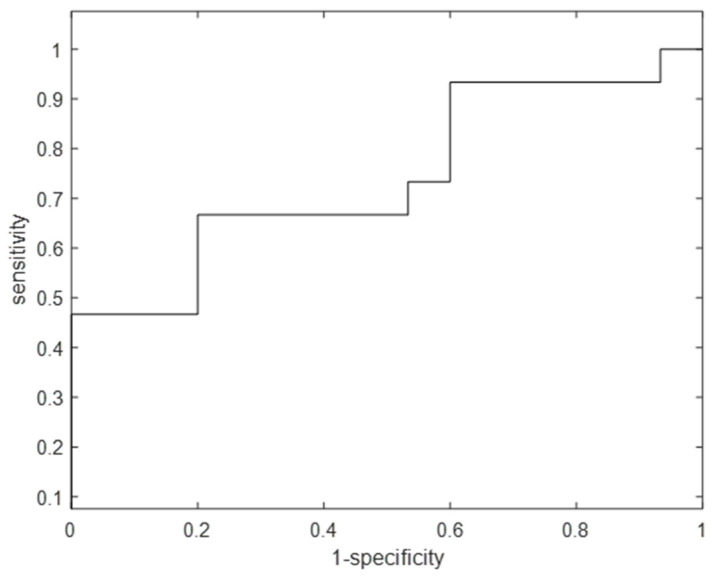
Receiver Operating Characteristics curve for the test set of the random forest model comparing individuals before and after ICS treatment using 20 discriminatory VOCs. The sensitivity and specificity were found to be 73% and 67%, respectively, with an area under the curve equal to 0.72. ICS: inhaled corticosteroids; VOCs: volatile organic compounds.

**Figure 3 jcm-11-05160-f003:**
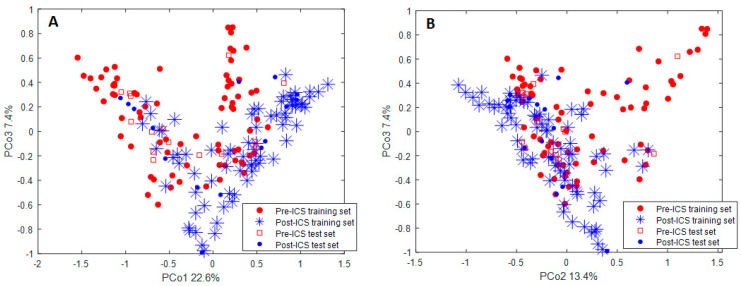
Principal Coordinate Analysis score plot for (**A**) PCo1 and PCo3, and (**B**) PCo2 and PCo3 on a proximity matrix obtained from the Random Forest model performed on the training set and 20 of the most discriminatory volatile organic compounds. The test set was projected into the space defined by the training samples. Every point belongs to a single breath fingerprint (blue stars: post-ICS training set; red circles: pre-ICS training set; blue circles: post-ICS test set; red squares: pre-ICS test set). The separation is observed on Principle Component 1, explaining 22.6% of the variance. PCo: Principal Coordinate; ICS: inhaled corticosteroids.

**Table 1 jcm-11-05160-t001:** Baseline characteristics of responders and non-responders.

	Responders	Non-Responders
Study population (*N*)	46	43
Age (years), mean (SD)	3.2 (0.7)	3.4 (0.6)
Sex: male/female, *n*/*n*	20/26	25/18
Wheezing episodes, *n* (IQR)	8 (3–8)	8 (3–8)
Atopy, *n*/*n* (%)	12/45 (27)	9/41 (22)
Allergic rhinitis, *n*/*n* (%)	3/45 (7)	3/42 (7)
Eczema, *n*/*n* (%)	15/44 (34)	12/43 (28)
Positive API, *n* (%)	5/44 (11)	4/43 (9)
Inhaled corticosteroids, *n*/*n* (%)	12/45 (27)	9/42 (21)
Short acting β_2_-agonists, *n*/*n* (%)	23/46 (50)	19/42 (45)
Long acting β_2_-agonists, *n*/*n* (%)	1/46 (2)	0/42 (0)
Total symptom score, mean (SD)	24.7 (5.2)	26.4 (3.1)
Airway resistance (kPa s/L), mean (SD) *	1.6 (0.4)	1.4 (0.3)

Atopy is defined as a positive Phadiatop Infant test^®^. SD: standard deviation; IQR: interquartile range; API: stringent asthma predictive index. * *p* = 0.005 between responders and non-responders (Mann–Whitney U test). All other characteristics were statistically not significant (*p* > 0.05).

**Table 2 jcm-11-05160-t002:** Putative identification of the top 5 volatile organic compounds contributing to differentiation of individuals before and after treatment with ICS. ↓ indicates reduced level of a VOC in post-ICS individuals in reference to pre-ICS; ↑ indicates increased level of a VOC in post-ICS individuals in reference to pre-ICS.

Nr.	Compound Name	Concentration Changes in Post-ICS
1	Branched C_11_H_24_	↓
2	Butanal	↓
3	Octanal	↓
4	Acetic acid (ester)	↑
5	Methylated pentane	↓

ICS: inhaled corticosteroids; VOC: volatile organic compound.

## Data Availability

The data presented in this study are available from the corresponding author upon request.

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
