# Peer review of "Exhaled Breath Analysis for Investigating the Use of Inhaled Corticosteroids and Corticosteroid Responsiveness in Wheezing Preschool Children"

_jcm, 2022, doi:10.3390/jcm11175160_

Round 1

Reviewer 1 Report

This study determined the influence of inhaled corticosteroids (ICS) on exhaled breath profiles of wheezing preschool children. Moreover, the authors tried to determine whether exhaled VOCs could predict a clinical steroid response in wheezing preschool children. The topic is interesting because of the need to develop a non-invasive test that predicts steroid responsiveness in wheezing preschool children. The authors showed that exhaled volatile organic compounds (VOCs) in wheezing preschool children were influenced by ICS treatment. They found that a set of 20 VOCs could best discriminate between measurements before and after ICS treatment with determining sensitivity and specificity. Moreover, the top 5 discriminative VOCs were identified.

The article is well written and there is a good coherence between the different parts. The used technique of exhaled breath analysis was described and references that include more details were cited in part Materials and Methods.

Because of some methodological limitations, the authors failed to fully meet the objectives of the study. However, their findings may be useful for future studies in the purpose to standardize the method of exhaled breath analysis and to justify its usefulness in clinical practice.

Author Response

We would like to thank the reviewer for reviewing our manuscript and the positive comments.

Reviewer 2 Report

In this study Bannier and co-workers we investigated the influence of inhaled corticosteroids (ICS) on exhaled volatile organic compounds (VOCs) of wheezing preschool 20 children. A set of 20 VOCs could best discriminate between measurements before and after ICS treatment, with a sensitivity of 73%, and specificity of 67% (area under ROC curve = 0.72). The authors conclude that ICS significantly altered the exhaled breath profiles of wheezing preschool children, irrespective of clinical treatment response. Furthermore, exhaled VOCs were capable to determine corticosteroid responsiveness in wheezing preschool children.

There are some points to be clarified about this manuscript which are as follows:

Abstract:

It is not clear with which method the authors evaluate the VOCs (it can be seen in the middle of the manuscript). Furthermore in the abstract the authors describe which VOCs have been evaluated (i.e. “Most discriminative VOCs were branched C11H24, butanal, octanal, acetic acid, and methylated pentane. Other VOCs predominantly included alkanes”). But there is no trace in the methods. The authors could also clarify this point in the methods.

Methods:

It's a little confusing. The authors show “50 healthy controls,” but these then get lost in the manuscript. If they do not return to the analysis of this manuscript, they can be put online or even removed.

Results:

It might be useful to consider a table describing in more detail the cohort at the "end" of follow-up.

In table 1 it would be important to report a column with the significance p values.

The authors affirm that “A set of 20 VOCs was selected as being most discriminative”. What are these20 VOCs compounds?

In figure 2 the y-axis is incomprehensible. It would be interesting to see the multivariate model (line 272).

Author Response

We would like to thank the reviewer for the positive comments and constructive recommendations to improve our manuscript.

In the following point-by-point response we address the comments.

Abstract:

It is not clear with which method the authors evaluate the VOCs (it can be seen in the middle of the manuscript). 

Reply:

To clarify the abstract, we added the sentence “Exhaled VOCs were measured by GC-tof-MS” in line 24.

Furthermore in the abstract the authors describe which VOCs have been evaluated (i.e. “Most discriminative VOCs were branched C11H24, butanal, octanal, acetic acid, and methylated pentane. Other VOCs predominantly included alkanes”). But there is no trace in the methods. The authors could also clarify this point in the methods.

Reply:

The identification of VOCs is described in lines 167-9: “Lastly, the set of discriminating VOCs was putatively identified using the NIST library (National Institute of Standards and Technology) in combination with expert interpretation.” In the Results section this is again clarified in lines 244-5.

Methods:

It's a little confusing. The authors show “50 healthy controls,” but these then get lost in the manuscript. If they do not return to the analysis of this manuscript, they can be put online or even removed.

Reply:

We agree that the 50 healthy controls are confusing. As these subjects were not included in our analyses, we have removed this detail (line 73).

Results:

It might be useful to consider a table describing in more detail the cohort at the "end" of follow-up.

Reply:

The ADEM cohort was followed until age 6 years, when a definitive diagnosis of asthma was made. However, the primary aims of the present paper is the influence of ICS on exhaled VOCs, and the usefulness of exhaled VOCs to predict a steroid response. These were both determined at preschool age. We only used the final diagnosis of asthma or transient wheeze as a main factor in the rMANOVA analysis. We therefore think that a table describing the characteristics of our cohort at age 6 would divert the attention from our primary aims and findings. 

In table 1 it would be important to report a column with the significance p values.

Reply:

In the revised version, we now clarify that all other baseline characteristics (beside airway resistance) were statistically not significant (p > 0.05), we added this to lines 186-7.

The authors affirm that “A set of 20 VOCs was selected as being most discriminative”. What are these 20 VOCs compounds?

Reply:

The discriminatory multivariate model allowed us to select the set of 20 VOCs. The most contributing VOCs were identified and included in the manuscript (see Table 2). The remaining compounds were only identified on the family level, where it is specified in the manuscript that they belong to terpenes (2), alkanes (7) and branched alkanes (4). Moreover 2 were unidentified. Please take into account that mass spectra that were obtained in the study are not high resolution, making it impossible to make distinction between some of the compounds, such as terpenes, which show a very similar mass spectrum. The same applies to branched alkanes. The exact structure of the compound is impossible to give because there are many isomers which give an identical mass spectrum. If run without a chemical standard, these are impossible to identify.

In figure 2 the y-axis is incomprehensible.

Reply:

Our sincere apologies for this inconvenience due to inserting the figure wrongly in the Word document. The y-axis states ‘sensitivity’ with a scale from 0 to 1.

It would be interesting to see the multivariate model (line 272).

Reply:

We do not really understand this question. We cannot provide the coefficients of the model as this is not the type of the technique we used to evaluate the ICS response. We have used a MANOVA model to test the significance of the different effects: response to treatment for later diagnosed transient wheezing children and true asthma children. We have tested each of the factor (i.e. response to therapy and disease type) and the interaction between them. We have provided the significance level of each main factor and the interaction factor. We can provide other statistical measures here like F-statistics or error but in our view, this is not very helpful.